# Clinical Spasticity Assessment Assisted by Machine Learning Methods and Rule-Based Decision

**DOI:** 10.3390/diagnostics13040739

**Published:** 2023-02-15

**Authors:** Jingye Yee, Cheng Yee Low, Natiara Mohamad Hashim, Noor Ayuni Che Zakaria, Khairunnisa Johar, Nurul Atiqah Othman, Hock Hung Chieng, Fazah Akhtar Hanapiah

**Affiliations:** 1Faculty of Mechanical and Manufacturing Engineering, Universiti Tun Hussein Onn Malaysia, Parit Raja 86400, Malaysia; 2Faculty of Medicine, Universiti Teknologi MARA, Sungai Buloh 47000, Malaysia; 3College of Engineering, Universiti Teknologi MARA, Shah Alam 40450, Malaysia; 4Department of Computing and Information Technology, Tunku Abdul Rahman University of Management and Technology, Kampar 31900, Malaysia; 5Daehan Rehabilitation Hospital Putrajaya, Putrajaya 62502, Malaysia

**Keywords:** spasticity, Modified Ashworth Scale, machine learning, medical expert system

## Abstract

The Modified Ashworth Scale (MAS) is commonly used to assess spasticity in clinics. The qualitative description of MAS has resulted in ambiguity during spasticity assessment. This work supports spasticity assessment by providing measurement data acquired from wireless wearable sensors, i.e., goniometers, myometers, and surface electromyography sensors. Based on in-depth discussions with consultant rehabilitation physicians, eight (8) kinematic, six (6) kinetic, and four (4) physiological features were extracted from the collected clinical data from fifty (50) subjects. These features were used to train and evaluate the conventional machine learning classifiers, including but not limited to Support Vector Machine (SVM) and Random Forest (RF). Subsequently, a spasticity classification approach combining the decision-making logic of the consultant rehabilitation physicians, SVM, and RF was developed. The empirical results on the unknown test set show that the proposed Logical–SVM–RF classifier outperforms each individual classifier, reporting an accuracy of 91% compared to 56–81% achieved by SVM and RF. A data-driven diagnosis decision contributing to interrater reliability is enabled via the availability of quantitative clinical data and a MAS prediction.

## 1. Introduction

Spasticity is a common phenomenon seen in neurological disorders. Clinically, spasticity manifests as an increased resistance offered by muscles to passive stretching or lengthening [1]. It is a velocity-dependent increase in muscle tone caused by the increased excitability of the muscle stretch reflex. It is common among patients with brain injury, cerebral palsy, multiple sclerosis, spinal cord injury, and stroke.

Spasticity has a broad impact on the lives of patients and their families. Spasticity can cause discomfort and stiffness, while spasms can be annoying and painful and may interfere with function [2]. Not only mobility or physical activities of patients can be affected, but the ongoing presence of spasticity and spasms can also have an emotional impact. For instance, the localisation of spasticity in both legs or the right arm can produce a significant impact on the ‘Need for Assistance/Positioning’ and ‘Social Embarrassment’ [3].

The clinical management of spasticity is linked to the severity of the spasticity. However, an accurate and reliable severity assessment is a challenging area [2]. Research is ongoing into different assessment strategies, including gait analysis and biomechanical, neurophysiological, and clinical measurements. The Modified Ashworth Scale (MAS) [4] is one of the commonly used clinical scales to assess spasticity, and MAS is also the most cited scale in the literature [5]. The MAS tests resistance against passive movement within a range of motion (ROM) about a joint with varying degrees of velocity. MAS scores range from 0 to 4, with six grades (0, 1, 1+, 2, 3, 4). The description of the MAS is visualised in Table 1.

### Related Works

Some relevant studies were conducted prior to this study comprising different demography, number of recruited subjects, types of collected parameters, extracted features, classification approaches, and MAS levels. An Automatic Muscle Spasticity Assessment System (AMSAS) [6] for assisted diagnosis was developed for the MAS clinical scale of 0, 1, and 2, while Levels 1+, 3, and 4 were excluded from the study. The system was developed with elbow torque and angle data collected from 25 subjects, and three machine learning classifiers were evaluated in the study. The seven features used for system training were mechanical features such as work done, torque, and stiffness.

Multilayer Perceptron (MLP) was used in an attempt to imitate the clinical assessment of spasticity with MAS [7]. Thirty-four subjects were recruited for the clinical data collection, and nine biomechanical parameters were used as feature inputs for the MLP training. However, it only covered the assessment of MAS Level 0 to Level 3. The training and deployment of MLP consume more computational resources and require a larger dataset to ensure a decent classifier is trained.

Another regression-based assessment system [8] with surface electromyography (EMG) and inertial sensors was developed for a similar purpose. Sixteen subjects with spasticity were included in the study, and a fusion model was developed for the assessment purpose. No MAS Level 4 subject was included in the study, and the MAS clinical scale was treated as a continuous scale with MAS Level 1+ assumed as MAS 1.5 (between MAS Level 1 and MAS Level 2).

There was also an attempt to assess spasticity based on voluntary movement of the spastic arm [9] instead of the passive movement. Machine learning was used to evaluate the degree of spasm with the sensor data from surface EMG and Inertial Measurement Unit (IMU). Thirteen subjects were recruited (nine spastic and four non-spastic). Four classification models were tested in the study. MAS Levels 3 and 4 were excluded from the study, as subjects with MAS Levels 3 and 4 are not able to perform active movement due to pain or limited mobility.

Nonetheless, all the systems mentioned either excluded certain levels from the MAS clinical scale or proposed an entirely new scale in the assessment process. This deviates from the diagnosis procedure in a clinical setting and is not able to assist the physician in their decision-making process. This study proposes a fusion model of simple logical decision and supervised learning classifiers to assist the clinical assessment of spasticity based on the MAS clinical scale. The logical decision rule and the classifiers’ training features are determined from the direct consultation and close engagement with the consultant rehabilitation physicians to closely imitate the physicians’ decision-making process in clinical settings.

## 2. Materials and Methods

### 2.1. Clinical Data Collection

Ethical approval was acquired from the Research Ethics Committee of the Universiti Teknologi MARA (UiTM) for the clinical data collection sessions. Patients suffering from upper limb spasticity, specifically on their flexor muscle, were recruited for the clinical data collection. The patients were recruited from the UiTM Private Specialist Centre in Sungai Buloh in Malaysia.

The inclusion criteria for patients were:Presence of any central nervous system pathology.Possess good cognitive function determined by Mini-Mental State Examination (MMSE) with a score ≤24.

Meanwhile, the exclusion criteria for patients were:Presence of elbow joint or forearm pathology secondary to the non-neurological cause.Presence of elbow joint contracture secondary to bone pathology.The process of recruiting the patients up to the appointment and engagement of patients is summarised in Figure 1.

The process began with identifying potential patients based on the available patient database. The rehabilitation physicians involved in this study are two practicing clinicians and academicians in the field of rehabilitation medicine. The rehabilitation physicians either embedded the data collection appointment during a normal clinical consultation session with the patients or purposely invited the patients for a data collection session. With the patient’s consent, an appointment for a data collection session was made by deciding the date and venue. A few days before the data collection session, a reminder call was made to the patients regarding the appointment. Finally, the patient arrived at the venue in the presence of a rehabilitation physician. The patients and their caretakers were briefed about the entire data collection process. The physicians checked the patients’ suitability to participate in the session based on a set of inclusion and exclusion criteria. The consent and signature of the patients (or their caretakers) was acquired before the data collection process.

#### 2.1.1. Data Acquisition System

Quantitative spasticity data are vital for data-driven classifier training. The crucial parameters are the elbow angular motion, the resisting force (or torque) of the patient’s elbow, and the surface electromyography (sEMG) of the bicep of the spastic arm. Three sensors are needed, i.e., a goniometer (elbow angular motion), a myometer (resisting force/torque), and a sEMG sensor. All sensors involved in this study are products of Biometrics Ltd. (Newport, UK), a company that specialises in sensors for biomedical and engineering purposes. The specifications of the sensors are tabulated in Table 2.

The sensors are then integrated into a complete data acquisition system with the connector and the data acquisition terminal. Essential considerations in developing the data acquisition system include the actual clinical settings, the ease in strapping the sensors onto the body parts, and ergonomics. The ease in setting up sensors positively influences the willingness of physicians to adopt the technology and the efficiency of carrying out the data collection with the sensors. Thus, all the sensors used in this study were either wireless by design or connected to a wireless interface. The sensors for the data acquisition system are shown in Figure 2.

The signals acquired by the wireless sensors were transmitted to the data acquisition terminal (local computer) through a dongle wireless transceiver. The DataLITE data acquisition software (ver. 10.28) of Biometrics Ltd. (Newport, UK) was installed on the local computer. All the signals were sent to the DataLITE or a third-party software through the Application Programming Interface (API) of the DataLITE software. The connectivity between the sensors and the data acquisition device is illustrated in Figure 3.

The data were collected at a frequency of 2000 Hz for the surface EMG, while the signals of elbow angular motion and resisting force were collected at a frequency of 1000 Hz throughout the extension of the spastic arm. The received signals were then saved in the local computer for further exploration and analysis.

#### 2.1.2. Data Collection Process

The data collection process consists of three major stages: the pre-assessment stage, the assessment stage, and the post-assessment stage. The steps of the data collection process are shown in Figure 4.

The pre-assessment phase was the setting up of a data acquisition system (sensors and video recording equipment) and data acquisition terminal (laptop computer and software) as preparation for the assessment phase. This phase is essential in ensuring proper data collection and providing a reference for further analysis. The sensors involved were the sEMG sensor, the twin-axis electrogoniometer, and the myometer. All the sensors are commercial sensors supplied by Biometrics Ltd. The sEMG sensor and the twin-axis electrogoniometer are wireless. The myometer is not wireless by design; a wireless adaptor provided the wireless connection with the data acquisition terminal.

The sensors were attached to different parts of the elbow to obtain the corresponding data for the data-driven classifier development: the sEMG sensor at the muscle belly of patient’s bicep, the twin-axis electrogoniometer at the dorsal surface (one end at the upper arm, another end at the forearm), and the myometer strapped onto the hand of clinician and held between the hand of clinician and the wrist of the patient. The sEMG sensor was attached to the skin on top of the bicep muscle belly with EMG sensor tape. The twin-axis electrogoniometer was attached to the dorsal surface of the patient’s upper limb with a goniometer tape. Both tapes mentioned are hypoallergenic and latex-free [10], so they are safe for clinical use on human skin.

The next step was setting up the data acquisition software. DataLITE is the data acquisition software provided by Biometrics Ltd., and its wireless connection with the sensors can be established through the DataLITE dongle wireless transceiver. The DataLITE dongle wireless transceiver was plugged into a computer to establish the connection between the DataLITE sensors and the computer.

The assessment phase is the phase when the sensors’ data were collected during the movement of the patient’s spastic arm. The clinicians adopted a standard protocol in the evaluation of spasticity. The protocol involved determining the patient’s spastic arm, followed by assessing the spasticity severity level based on the MAS. The ROM is the range of movement that the patient’s arm can move from the fully flexed position to the fully extended position. The assessment phase can be divided into three exercises: passive slow stretches applied on affected elbow flexors (repeated three times consecutively), passive fast stretch applied on affected elbow flexors (repeated three times consecutively), and the assessment of spasticity severity level based on MAS. The assessment phase is shown in Figure 5.

With the completion of the assessment phase, the data from the six stretches were acquired by the sensors and temporarily saved in the DataLITE software (ver. 10.28). The file was then be saved on the local computer in a Log Data file format with a .LOG extension. All the collected data in the .log format were used for further exploration and analysis, with the goal of training a machine learning algorithm for computer-assisted diagnosis of upper limb spasticity.

#### 2.1.3. Data Pool

Fifty patients were recruited, with 1 to 3 trials conducted with each patient. However, 3 of them were discarded due to their failure to fulfil the second inclusion criterion (passing the MMSE). The mean age of the patients was 45.26 ± 19.99. There were 96 trials in the data pools with various MAS levels. The full table of the recruited subjects with corresponding MAS ratings and details is attached as Table A1 (Appendix A). The distribution of the MAS level of the subjects is summarised in Table 3. The complete clinical dataset will be published online with the publication of this study.

### 2.2. Data Preprocessing

The data collected was now in its raw and primitive form. The next step was to pass the raw data through the data preprocessing pipeline to turn the data into extracted features. The raw data included elbow angle (in the separate x-axis and y-axis), elbow resisting force, and surface electromyography (sEMG). The types of raw data collected and their corresponding details are shown in Table 4.

Data must be preprocessed before they can be used for the training of the classifier. For this specific use case of upper limb spasticity diagnosis, a data preprocessing flow (Figure 6) was determined, including data integration, data segmentation, and data cleaning and filtering, followed by features extraction.

#### 2.2.1. Data Integration

The sensors used in this study were supplied by the same company, thus, the sensors’ data were synchronised with the DataLITE software. Hence, the data integration was only required for the elbow angle. Since a twin-axis electrogoniometer was used, the elbow angle data was obtained in a pairing value for the *x* and *y* planes. Thus, the pairing values were integrated as one value by finding the resultant elbow angle using Equation (1) below:(1)$$Elbow\;Angle=\sqrt{\theta_{x}{}^{2}+\theta_{y}{}^{2}}$$

#### 2.2.2. Data Segmentation

The data acquired in a single session consisted of six stretches: three slow stretches followed by three fast stretches. The slow stretches provided basic information such as the full range of motion (ROM) of the spastic arm before entering the fast stretches phase. Characteristics of spasticity, such as catch, can only be observed during fast stretches. Hence, it is essential to distinguish between fast stretches and slow stretches and isolate the fast stretches for features extraction. The segmentation process of the samples was automated with a Python script to reduce the unnecessary workload of manual segmentation. To determine the time windows of the stretches, an algorithm was written to detect the local minima and the local maxima of the elbow angular motion, and the result is shown in Figure 7. The obtained indices of the starting and ending positions of the stretches were used for segmentation of the data window of each stretch for the elbow resisting force and sEMG.

#### 2.2.3. Data Cleaning and Filtering

The next step was data cleaning and filtering, which transformed the raw data into a ready form to extract valuable information for further analysis. The filters utilised for this process include median filter, mean filter, fix zero data levelling, zero mean value data levelling, and Butterworth filter. The filters are applied for a few purposes: (1) to filter the noise or remove outliers; (2) to further remove zero-error; and (3) to prepare the time-series data for valuable information extraction. The cleaning and filtering processes applied to the parameters are detailed in Table 5.

The differences after the filtering and cleaning processes of the elbow resistance and the sEMG are illustrated in Figure 8 and Figure 9. The graph for the elbow angle is not shown as the differences are not visible in a graph of small scale.

#### 2.2.4. Features Extraction

The accuracy and precision of the data-driven classifier can be improved if the distinguishing features are fed into the model. An interview was carried out with two consultant rehabilitation physicians and one physiotherapist from the Faculty of Medicine of the Universiti Teknologi MARA (UiTM). The consultation aimed to identify essential clues of how the clinicians diagnose the severity level of the upper limb spasticity. From the interview, several parameters and characteristics were identified as “important features” for the classification. The extracted features can be categorised into three divisions: the kinematic features, the kinetic features, and the physiological features.

The kinematic features are the features related to the angular position, velocity, or acceleration of the elbow joint. The term “catch” (also known as “jerk”) mentioned below is defined as the instance when the elbow experiences the maximum deceleration, thus resulting in a jerk-like movement during the passive stretch of the elbow. All the kinematics-related features are as follows:Range of Motion (ROM): ROM is the maximum range of motion that the elbow can move between its fully flexed position and fully extended position;Angle of Catch (θ_catch_): θ_catch_ is the elbow angle which coincides with the elbow catch during the passive stretching;Ratio of Angle of Catch to Range of Motion: The ratio of the catch angle to the ROM;Angle of Maximum sEMG (θ_MsEMG_): θ_MsEMG_ is the elbow angle which coincides with the instance of maximum sEMG value during the passive stretching;Ratio of Angle of Maximum sEMG to Range of Motion: The ratio of the angle of maximum sEMG to the ROM;Angle of Maximum Elbow Force (θ_MForce_): θ_MForce_ is the elbow angle which coincides with the instance of maximum elbow resisting force value during the passive stretching;Ratio of Angle of Maximum Elbow Force to Range of Motion: This is the ratio of the angle of maximum elbow resisting force to the ROM;Maximum Angular Velocity (ω_max_): ω_max_ is the maximum value of the elbow angular velocity throughout the elbow passive stretching process.

The kinetics features are the features related to the elbow resisting forces and their relationships. All the kinetics-related features are as follows:Ratio of Elbow Force at Catch to Initial Elbow Force: This ratio is to analyse the fold of increment of elbow resisting force at the catch compared to the initial elbow force;Normalised Increment of Elbow Force at Catch: This feature provides information about the increment of elbow force at catch compared to the initial elbow force. The value is then normalised as the combinations of different patients and clinicians of distinguished gender, age, and the background induce different magnitude of forces;Normalised Average Elbow Force: This feature provides information about average elbow force throughout the elbow extension process. The value is normalised to produce a force value relative to the maximum force during the engagement;Normalised Average Elbow Force after Catch: This feature provides information about average elbow force throughout the remaining elbow extension process after the catch. The value is normalised to produce a force value relative to the maximum force during the engagement;Average Slope of Elbow Force after Maximum Elbow Force: This feature shows the average rate of change in elbow force after the occurrence of the maximum elbow force;Average Slope of Elbow Force after Catch: This feature shows the average rate of change in elbow force after the occurrence of the catch.

Physiological features refer to the features related to the muscle and its activity. All the physiological-related features are as follows:Ratio of Elbow Force at Catch to Initial Elbow Force: This ratio is to analyse the fold of increment of elbow resisting force at catch compared to the initial elbow force;Normalised sEMG at Catch (Normalised sEMG_catch_): This feature is the value of sEMG at the occurrence of catch. The value is then normalised by dividing it with the maximum sEMG value to represent it as a relative value instead of an absolute magnitude;Normalised Average sEMG after Catch: This feature is the average value of sEMG after the occurrence of catch. The value is then normalised by dividing it with the maximum sEMG value to represent it as a relative value instead of an absolute magnitude;Normalised Average Slope of sEMG after Catch: This feature is the average value of the change in sEMG after the occurrence of catch. The value is then normalised by dividing it with the maximum sEMG value to represent it as a relative value instead of an absolute magnitude.

### 2.3. Dataset Preparation

#### 2.3.1. Removal of Slow Stretches

The features were extracted from both the slow stretches and fast stretches. A total of 96 trials were conducted on the patients, with each trial providing data of three slow stretches and three fast stretches, bringing the total number of stretches to 540. However, only the 270 fast stretches were used for the training, while the 270 slow stretches were excluded.

The dataset contains samples of the spastic elbow at different MAS levels. As shown in Table 4, there are 72 samples for MAS Level 0, 84 samples for MAS Level 1, 66 samples for MAS Level 1+, 27 samples for MAS Level 2, 21 samples for MAS Level 3, and 6 samples for MAS Level 4.

As the dataset is considerably small in scale, the splitting ratio of the train set to test set used was 90:10. The stratified splitting was used in the train–test splitting process, which was applied by first dividing the datasets into strata (homogeneous subgroups) and sampling each stratum to ensure the test set was representative of the dataset’s population [11]. The number count of the different MAS level strata within the train and test sets is shown in Table 6. The train set was utilised in the training process, while the test set was kept aside for the model validation. It should be noted that the test set remained unseen by the computer during the classifier training process.

#### 2.3.2. Data Normalisation

After the train–test sets splitting, the values of the features had to be normalised to prevent the domination of large numeric attributes over the smaller numeric attributes [12]. Additionally, the normalisation within the data of each patient itself can eliminate the issue of different reaction values across different patients due to their individual strength and stature. The normalisation was carried out after the train–test splitting so that it could resemble the actual application in the clinical setting later, as the features extracted from the newly acquired data were normalised on their own before they were fed for MAS level classification. Normalisation was carried out by removing the mean value and scaling to unit variance.

### 2.4. Experimental Setup

#### 2.4.1. Upsampling

The sample size of different MAS levels in the train set was highly imbalanced due to the scarcity of patients with more severe upper limb spasticity conditions (higher MAS levels) in the population. Upsampling is applied to the classes of a smaller sample size to provide more representation in training. This approach avoids the classes of smaller sample sizes being overshadowed by the classes of larger sample sizes.

Two upsampling methods were used in this study, i.e., the Random Oversampling (ROS) and the Synthetic Minority Over-sampling Technique (SMOTE) [13]. ROS is a primitive upsampling method that randomly duplicates the samples of the particular MAS levels so that the samples in the MAS level will be the same as the other MAS levels with more samples. SMOTE employs median and nearest-neighbour computation [13] to produce new data points, which is not similar to the existing samples.

#### 2.4.2. Hyperparameters Fine-Tuning

After the upsampling, the hyperparameters of classifiers were fine-tuned to produce better classification results. The grid search method was applied to identify the best parameters for the hyperparameter tuning process. After the grid search method returned the best hyperparameter values based on the specified scoring method, manual fine-tuning was further applied to adjust the hyperparameters to a more realistic value and prevent data overfitting. The grid search value ranges for the different hyperparameters are tabulated in Table 7.

#### 2.4.3. Classifiers Training and Performance

After fine-tuning the hyperparameter value, each classifier was trained with the defined hyperparameters with a 10-fold cross-evaluation to obtain the training performance result. The machine learning algorithms deployed and compared are the Gaussian Naïve Bayes [14], Decision Trees [15], Random Forest [16], XGBoost [17], and Support Vector Machine (SVM) [18]. All the algorithms of the classifiers in this study are directly called from the Scikit-learn library [19]. The Scikit-learn package provides simple and efficient tools for predictive data analysis and is based upon the other libraries such as NumPy, SciPy, and Matplotlib.

The performance of the classifiers was exhibited on two occasions: once during the data-driven model training process with the 10-fold cross-evaluation process on the train set and once during the validation with the test set. A special note is that all the cases of MAS Level 4 were removed from the training of classifiers entirely. There are two reasons for the removal. First, the instances of MAS 4 were scarce in our samples, constituting only 6 out of 278 instances. The second factor is that the diagnosis of MAS Level 4 can be made with a straightforward logical decision: almost no range of motion (ROM ≈ 0). The logic rule was put into the diagnosis algorithm directly from the beginning, saving computation power to focus on the diagnosis of the other MAS levels.

#### 2.4.4. Performance Analysis on Train Set

During the classifier training process with the determined hyperparameters, the performance of the classifiers was measured with the scoring method of balanced accuracy [20], as shown in Equation (2).
(2)$$Balanced\;Accuracy\;\left(y,\hat{y},w\right)=\frac{1}{\sum {\hat{w}}_{i}}\underset{i}{\sum }1\left({\;\hat{y}}_{i}=y_{i}\right){\hat{w}}_{i}$$

where *ŵ_i_* = sample weight;and *y_i_* = value of the *i*-th sample.

In the case of the same sample size of all classes, the balanced accuracy was the same as accuracy. This is the case when upsampling or downsampling is applied to the whole dataset to have the same sample size for each class.

#### 2.4.5. Performance Analysis on Test Set

The performance of the classifiers on the test set is visualised with the confusion matrix and the normalised confusion matrix. The boxes along the diagonal from the top left corner to the bottom right corner of the confusion matrices are the instances where the classifiers made correct predictions of the data point. The higher the figure along this diagonal line, the better the performance of the classifier. The intensity of the blue colour is also used to represent the number on each box, with the darker blue colour representing a higher figure in a box. A confusion matrix with a dark blue colour along the diagonal line represents a good performance of the classifier.

Three performance metrics were used to evaluate the performance of the classifiers: Precision score (Equation (3)), Recall score (Equation (4)), and F-measure score (Equation (5)).
(3)$$Precision=\frac{TP}{TP+FP}$$
(4)$$Recall=\frac{TP}{TP+FN}$$
(5)$$F-Measure=\frac{2\cdot \left(Precision\cdot Recall\right)}{Precision+Recall}$$

where TP = True Positive;FP = False Positive;and FN = False Negative.

Another performance metric used for the evaluation is “Accuracy”, which is a relatively simple and primitive score. It is the percentage of correct classifications out of the total classification attempts, as shown in Equation (6).
(6)$$Accuracy=\frac{Correctly\;Classified\;Instances}{Total\;Classified\;Instances}$$

#### 2.4.6. Features Importance Analysis

Features importance analysis was carried out to identify the relative importance of the features in each classifier. This approach ensures the interpretability and transparency of the classifier in making the classifying decision known by the user of the system, especially in a critical setting such as in medical practice. In this study, SHapley Additive exPlanations (SHAP) [21] was used to measure the feature importance. The SHAP Python library was used to carry out the computations for that purpose.

## 3. Results

### 3.1. Hyperparameters of Classifiers

After the grid search and manual tuning, the optimised hyperparameters of the classifiers were determined. The hyperparameters for the ROS-upsampled and SMOTE-upsampled training samples for the different classifiers are listed in Table 8, Table 9, Table 10, Table 11 and Table 12. All the values in the tables in this section (if applicable) are shortened to four decimal points.

From Table 8, Table 9, Table 10, Table 11 and Table 12, it can be observed that the hyperparameters tuning on ROS-upsampled and SMOTE-upsampled training sets result in almost identical hyperparameters. There were only a few instances where the hyperparameters were slightly different in Decision Tree and Random Forest classifiers.

### 3.2. Performance of Classifiers

#### 3.2.1. Training Performance

The performance of the classifiers on the train set during the 10-fold cross-evaluation process is shown in Table 13, detailing the training results of both upsampling methods of ROS and SMOTE. The result shown is the mean performance score of the 10-fold cross-evaluation with the corresponding standard deviation.

From Table 13, it can be observed that there is minimal difference between the training results of ROS-upsampled and SMOTE-upsampled training sets. The ROS-upsampled training set fares slightly better than the SMOTE-upsampled set in most of the classifiers, especially for Decision Tree and Random Forest classifiers. It could be said that ROS is a better upsampling method for this particular use case. The reason could be that ROS randomly duplicates some existing data of the smaller classes to prevent overshadowing by larger classes, yet SMOTE expands the smaller classes by generating new data samples based on existing data, which could further blur the separating lines or planes between the classes. The only exception is in the Gaussian Naïve Bayes classifier, where the SMOTE-upsampled dataset has a slightly better training result than the ROS-upsampled dataset.

In terms of the training results, it can be observed that Random Forest and SVM perform equally well for ROS-upsampled datasets with similar standard deviations. Meanwhile, for the SMOTE-upsampled dataset, SVM outperforms all the other classifiers. Thus, the ROS-upsampled dataset was used further in the training of the classifiers in this study.

#### 3.2.2. Validation Performance

The performance of the classifiers on the test set was visualised with the confusion matrix and the normalised confusion matrix in Figure 10, Figure 11, Figure 12, Figure 13 and Figure 14. The boxes along the diagonal from the top left corner to the bottom right corner of the confusion matrices are the instances where the classifiers made correct predictions of the data point. The higher the figure along this diagonal line, the better the performance of the classifier. The intensity of the blue colour is also used to represent the number on each box, with the darker blue colour representing a higher figure in a box. A confusion matrix with a dark blue colour along the diagonal line represents a good performance of the classifier. Three performance metrics were used to evaluate the performance of the classifiers in this section: Precision, Recall, and F-measure score.

For the Gaussian Naïve Bayes classifier, the performance is visualised in Figure 10. The Gaussian Naïve Bayes classifier has a moderate performance and could be categorised as non-satisfactory for a medical setting. The classifier is especially poor in classifying MAS Level 0, with misclassification as MAS Levels 1, 2, and even 3.

The overall performance and detailed performance of the classifier for each MAS level are detailed in Table 14. The weighted average of the F-measure score is 0.56, and none of the F-measure scores were above 0.80.

For the Decision Tree classifier, the Scikit-learn package uses an optimised version of the Classification and Regression Trees (CART) algorithm, which constructs trees with binary rules that utilise the features and thresholds which return the most significant information gain at all the nodes [22]. The performance of the Decision Tree classifier is visualised in the confusion matrix and normalised confusion matrix in Figure 11 and detailed in Table 15.

The performance of the Decision Tree classifier is also moderate, with a weighted average F-measure score of 0.67. All MAS levels but MAS Level 3 have an F-measure score below 0.70.

For the Random Forest classifier, the performance is visualised in Figure 12 and Table 16. The performance of Random Forest was good, with F-measure scores of all MAS levels equal to or higher than 0.80. The weighted average F-measure score was 0.81.

For the XGBoost classifier, the performance is visualised in Figure 13 and detailed in Table 17. The XGBoost classifier performed well around MAS 1, 1+, and 2, with F-measure scores between 0.80 and 0.83. However, it performed less well for MAS Levels 0 and 3. The weighted average F-measure score was 0.75.

The performance of the SVM classifier is visualised in Figure 14 and Table 18. The performance of the SVM classifier was excellent for MAS Levels 1 (0.88) and 3 (1.00), while it performed fairly for MAS 0 and 1+. The classification prowess for MAS Level 2 was non-satisfactory. The weighted average F-measure score was 0.79.

As a summary, the Random Forest classifier had the best performance on the test set generally, with a weighted average F-measure score of 0.81, followed by SVM (0.79), XGBoost (0.75), Decision Tree (0.67), and Gaussian Naïve Bayes (0.56). The accuracy of the classifiers was similar to the weighted average F-measure scores. The general performance of the classifiers on all MAS levels is summarised in Figure 15.

#### 3.2.3. Features Importance Analysis

Figure 16, Figure 17, Figure 18 and Figure 19 contain the results of the SHAP feature importance analysis in the form of a bar chart. The y-axes are the features included in the classifier’s training, while the lengths of the bar charts represent the stacked importance of the features. The different colour segments in the stacked bar chart represent the features importance for different MAS levels.

In Figure 16, it can be seen that “Catch Angle” is a dominating feature for the decision-making of the Decision Tree classifier for both ROS- and SMOTE-upsampled datasets. It should be noted that the “Catch Angle: Range of Motion” feature has no influence in the ROS-upsampled dataset, as the limit of maximum features hyperparameter has made the classifier exclude the feature in all its decision-making processes.

For the Random Forest classifier, the top five dominating features are “Catch Angle”, “Range of Motion”, “Angle of Max Force”, “Normalised Average Slope of sEMG after Catch”, and “Force at Catch: Initial Force”. The “Range of Motion” feature was distinctive in classifying MAS 3 with a high average impact on both datasets.

In Figure 18, we can see one dominating feature in both bar charts, which is the “Catch Angle”. After that, there are three stand-out features in both bar charts, while the rest are comparatively less important. They are “Force at Catch: Initial Force”, “Range of Motion”, and “Normalised Average Slope of sEMG after Catch”.

For SVM, the main classifying feature is the “Force at Catch: Initial Force”. Medical doctors use this feature in deciding whether it is a non-spastic (MAS 0) or spastic (non-MAS 0) cases. Though the two datasets have different top five features, we can observe that “Range of Motion”, “Catch Angle”, and “Normalised Average sEMG after Catch” stay in the top five distinguishing features for both datasets.

In an overview of the SHAP analysis, the “Catch Angle” feature appeared as the top distinguishing feature for three classifiers (Decision Tree, Random Forest, and XGBoost), while also appearing in the top five distinguishing features for the SVM classifier. This is consistent with the primary consideration of the ULS diagnosis of the medical doctors we interviewed.

The other outstanding feature is “Range of Motion”. It appears in the top three positions for three classifiers, with the only exception of the Decision Tree. A closer look reveals that “Range of Motion” places a significant impact specifically on MAS Level 3 diagnosis, as MAS Level 3 has a minimal “Range of Motion” generally compared to the other MAS levels.

Another interesting observation is that “Force at Catch: Initial Force” is the top distinguishing feature in both the differently upsampled training sets for the SVM classifier, playing an essential role in diagnosing MAS Level 0 and 1+. The same feature appears in the top five features in the SHAP analysis of other classifiers. MAS Level 0 is non-spastic, and the minor increment in the Force at Catch compared to the Initial Force reflected that correctly. The MAS Level 1+ observation provides us with a new perspective on this feature in the diagnosis of ULS.

#### 3.2.4. Combined Classifier

In the manual diagnosis of upper limb spasticity, a few MAS levels are straightforward and apparent for the diagnosis. According to the clinicians, a spastic elbow of MAS Level 2 usually has a catch that occurs in the early phase of the stretch, which is generally interpreted as in the first half of the range of motion. However, the parameter is not definite and might vary from case to case. For MAS Level 3, the spastic arm is barely stretchable with constant force in action. According to the clinicians, the MAS Level 3 is considered a different MAS severity level with constant force throughout the stretch. The constant force results in the suppressed angular velocity of the spastic elbow during the stretch. In the case of MAS Level 4, the spastic arm is immobile, and its range of motion is severely limited. According to the clinicians, the range of motion is severely limited as if the elbow is not stretchable at all, and this is supported by the collected clinical data that show the maximum ROM to be 7.09°. The clinicians can quickly distinguish the case of MAS Level 4 by identifying the severely limited ROM during a slow stretch of the patient’s arm. The poor classification accuracy of MAS Level 4 due to the scarce data on MAS Level 4 cases can be overcome by adding a logic decision in the classifier. The logic decision can immediately classify the patient’s arm as MAS Level 4 if the elbow’s range of motion is smaller than 10°.

The different classifiers have their strength in distinguishing the different MAS Levels. Thus, the classifier can be enhanced with the combined processes of different logical decisions and classifiers. The detailed flowchart of the Logical–SVM–RF classifier is shown in Figure 20.

The advantages of different classifiers were utilised and combined into a flow of processes. The process begins with reading the range of motion (ROM) of the stretch. If the ROM is limited to under 10°, then the spastic elbow is classified as MAS Level 4 immediately, and the diagnosis result is displayed to the clinicians right away. If the ROM is not under 10°, the array of features are fed into the trained SVM classifier to detect whether it is a MAS 1 or 3. If the result returns non-MAS 1 or non-MAS 3, then the features array is fed into the trained Random Forest classifier to classify it as either MAS 0, MAS 1+, or MAS 2.

The proposed Logical–SVM–RF classifier was tested upon the test set. The performance of the proposed classifier in the prediction of MAS level of the spastic arm is visualised in the confusion matrix and normalised confusion matrix in Figure 21, and the performance metrics are detailed in Table 19.

Generally, there were a total of three misclassified cases, with one MAS 0 instance classified as MAS 3, one MAS 1 instance classified as MAS 0, and one MAS 1+ instance classified as MAS 2.

All but one case of each MAS Level 0, Level 1, and Level 1+ were correctly predicted. All MAS Levels 2, 3, and 4 were correctly predicted. Out of all the predicted MAS Level 0 cases, there was one case that was MAS 1. This means that the classifier is still imperfect in classifying between non-spastic (MAS 0) and spastic (non-MAS 0) arms.

Overall, the classifier correctly predicted 30 out of 33 cases with 91% accuracy. The weighted average F-measure was also 0.91. The Logical–SVM–RF classifier outperformed all the other individual classifiers in both Accuracy and the F-Measure score. The performance of the Logical–SVM–RF classifier is compared to the other individual classifiers in Figure 22.

In short, the Logical–SVM–RF classifier fares better than the individual classifiers working alone separately. The logical decision is first employed in the classifier, followed by the exploitation of the strengths of different classifiers to produce a combined classifier with better performance. The representation of the Logical–SVM–RF classifier is visualised in Figure 23.

## 4. Discussions

### 4.1. Logical–SVM–RF Classifier vs. Existing Assisted Diagnosis Method

The general approach to the development of upper limb spasticity classification models (or even for other purposes) is the training of individual classifiers/algorithms to recognise the different severity levels based on certain characteristics. This study proposes that the combination of decision-making logic and different classifiers could outperform the individual conventional machine learning algorithms. The comparison between this work and the existing works for the assisted diagnosis of upper limb spasticity is summarised in Table 20.

Due to the very different criteria and aspects espoused by the different works, it is an uphill task to compare the different works directly. The table above is a vague comparison of the different works to our best ability. For the first aspect, it can be seen that most studies (including this study) employ passive stretching (i.e., evaluator stretches the arm of subject) of the spastic arm, and only Chen et al. [9] employ active stretching (i.e., subject stretches own arm) of the spastic arm. However, MAS itself is a scale used for resistance measurement to passive movement [23]. Secondly, most of the studies excluded at least one MAS level from their classification effort, some citing no passive movement for MAS 4 [7]. Additionally, the studies which employed a regression method assumed that MAS is an interval scale (taking MAS 1+ as MAS 1.5), while MAS is actually an ordinal scale [4,5]. Thus, only this work covers the classification of all six MAS levels. Another important difference between this work and other works is the usage of the Logical–SVM–RF classifier, which is a combination of conventional classification models (SVM and RF) and the decision-making logic of clinicians. The other studies employ individual classifiers/regressors in their works, such as SVM, multilayer perceptron (MLP), single-/multi-variable linear regression, support vector regression (SVR), k-nearest-neighbour (KNN), and RF. As for the performance, the work in [9] showed a slightly higher F1 score compared to this study. However, it should be noted that the work only classified four MAS levels, compared to the six MAS levels in this work.

### 4.2. Towards Implementation in Clinical Setting

The current assessment method for upper limb spasticity based on the MAS is a manual and subjective approach. Timewise, the manual assessment method has a slight advantage over the assisted diagnosis method, as there is no need for setting up sensors and establishing a connection with the data acquisition system. However, the assisted diagnosis has the advantage of data transparency of the decision-making process and provides continuous data for further analysis and exploration. The classification model could provide a diagnosis reference to inexperienced clinicians to assist them in the decision-making process. A summary of the comparison is tabulated in Table 21.

With a proper graphical user interface (GUI) in place, the Logical–SVM–RF classification model could be combined with the sensors and data acquisition system to provide a user-friendly system to ease the work of the clinicians in a clinical setting.

### 4.3. Limitations

Developing a data-driven classifier assisted by machine learning for a medical expert system is an uphill task due to the difficulty in acquiring clinical data. The difficulty is magnified when dealing with less popular medical complications such as spasticity, researched in this study.

With a relatively small dataset, deep learning application becomes practically unfeasible, as training a deep learning model requires vast datasets. Thus, conventional classifiers were deployed in this study to train a data-driven classifier based on small datasets. To compensate for the small size of the dataset, different classifiers were combined into a logical decision flow to maximise the advantages of the different classifiers in detecting the different MAS levels. On top of that, a simple logical rule was employed in the flow (in determining MAS Level 4) based on the input from the clinicians. The combined Logical–SVM–RF classifier has shown better performance than the individual classifiers.

Another limitation of the small dataset is that the combined Logical–SVM–RF classifier could not be trained on the new dataset. All the datasets available were used as the train set to train the classifiers or were used as a test set to evaluate the performance of the classifiers. This result can be taken as the first step in testing out a combined classifier. The Logical–SVM–RF classifier will be tested with newly collected data in the future.

The limitation of this study is the scarcity of available patients with higher MAS levels, i.e., MAS Levels 3 and 4, which results in imbalanced clinical datasets, as the consent of patients is required.

In addition, the discrepancy in the assessment result of different evaluators on the same subject is an issue that must be solved. What makes the matter worse is that overstretching the subject’s spastic arm within a short period of time will temporarily diminish the severity level of the spasticity. Hence, it is justifiable to assign different MAS levels to the same subject even within the same data collection session. This also renders the action of conducting multiple trials on the same subject to increase the data pool relatively unfeasible.

## 5. Conclusions

This work presents an automated spasticity assessment supported by machine learning models based on the Modified Ashworth Scale (MAS). More specifically, a data-driven classifier that replicates the decision-making progression of clinicians is put forward. Essentially, our proposed model maps the quantitative data inputs of elbow motion, muscle signal, and muscle force into a predicted MAS scale ranging from Level 0 to 4. A Logical–SVM–RF classifier was assembled (logical rule with SVM and Random Forest algorithms) to improve classification performance with minimal computational power. It is noteworthy that our prediction is in line with the expectation of consultant rehabilitation physicians. We believe that the classifier’s performance can further be improved by collecting more comprehensive datasets, especially for MAS Levels 2 and 3. By providing quantitative clinical data and a MAS prediction, our work contributes towards reduced variability in the clinical assessment of spasticity using the Modified Ashworth Scale.

## Figures and Tables

**Figure 1 diagnostics-13-00739-f001:**
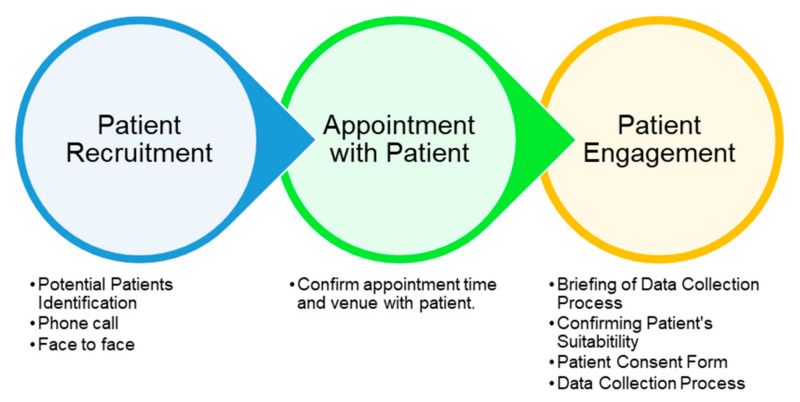
Process of Patient Recruitment and Engagement.

**Figure 2 diagnostics-13-00739-f002:**
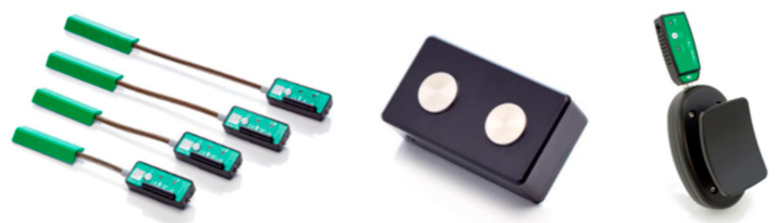
Sensors in data acquisition system: twin-axis goniometer (**left**), sEMG sensor (**middle**), and wireless-adapted myometer (**right**).

**Figure 3 diagnostics-13-00739-f003:**
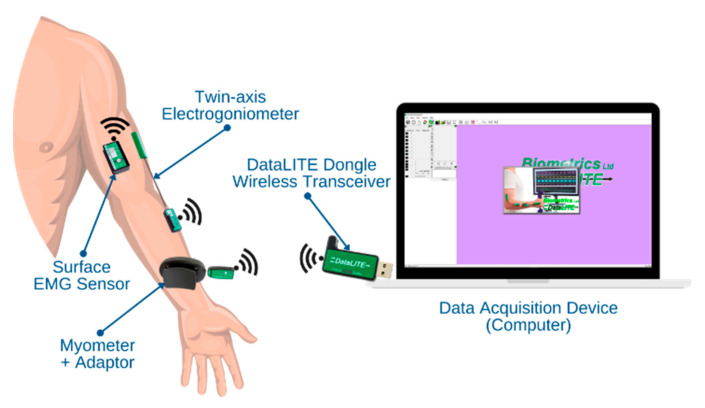
Connectivity between the DataLITE sensors and the computer.

**Figure 4 diagnostics-13-00739-f004:**
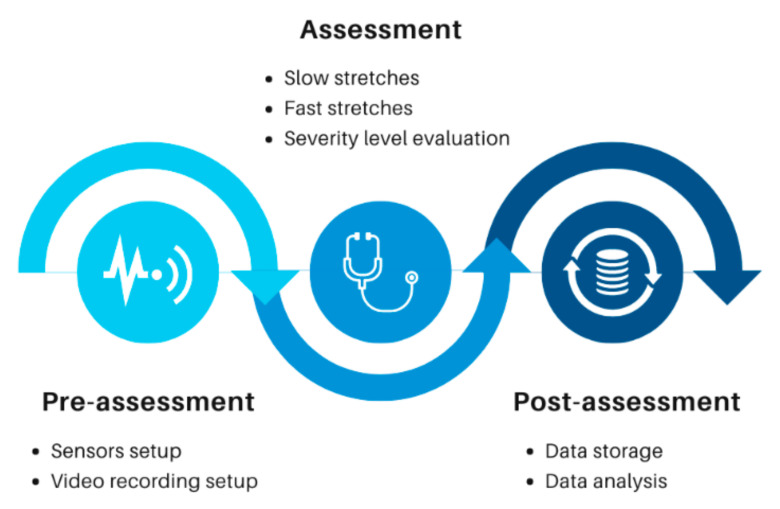
Flow chart showing the stages in clinical data collection.

**Figure 5 diagnostics-13-00739-f005:**
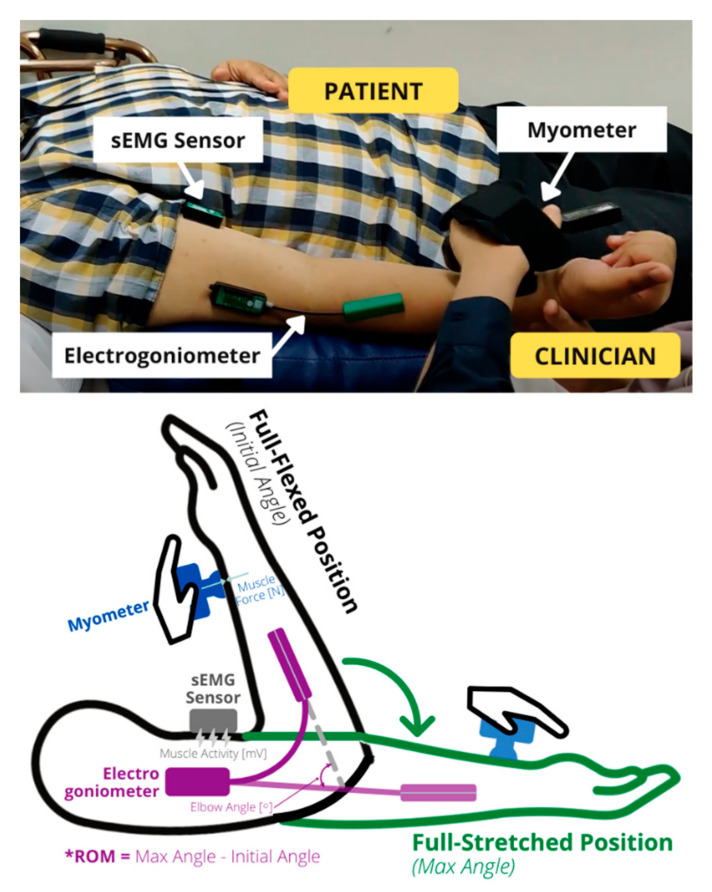
Illustration of the sensor’s positions and the passive stretch of the forearm. The forearm is stretched from a fully flexed position to a fully stretched position, and this difference in elbow angle is the range of motion (* ROM). The elbow angle, muscle force, and muscle signal are measured simultaneously during this motion.

**Figure 6 diagnostics-13-00739-f006:**
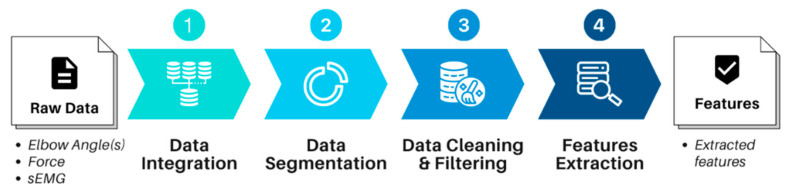
Data preprocessing stages to convert the raw data into extracted features.

**Figure 7 diagnostics-13-00739-f007:**
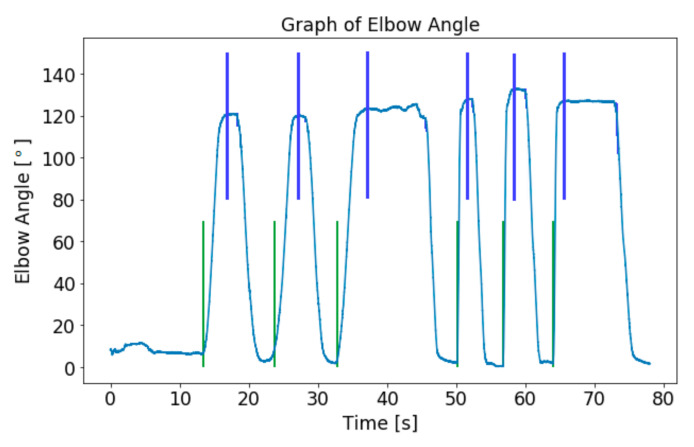
Detection of local minima (green line) and local maxima (blue line) of the elbow angle.

**Figure 8 diagnostics-13-00739-f008:**
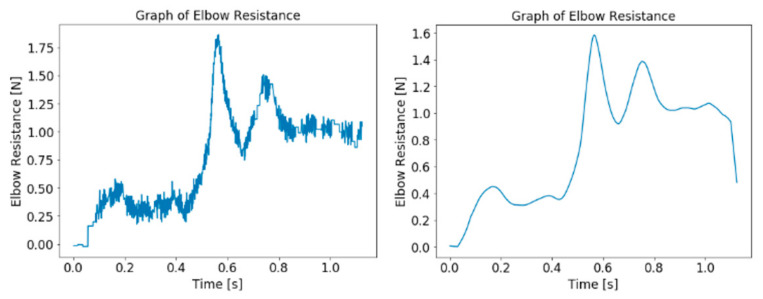
Graph of elbow resistance data before (**left**) and after (**right**) processing.

**Figure 9 diagnostics-13-00739-f009:**
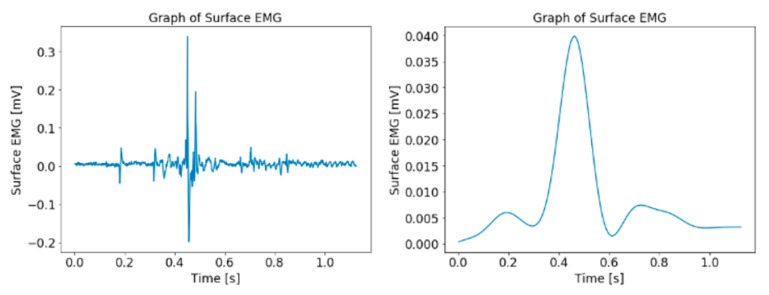
Graph of sEMG data before (**left**) and after (**right**) preprocessing.

**Figure 10 diagnostics-13-00739-f010:**
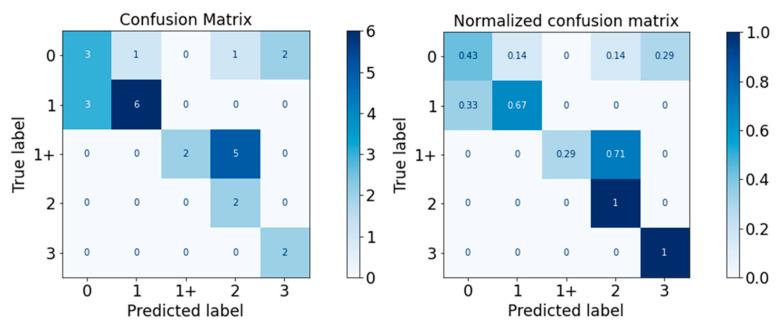
Confusion Matrix of Gaussian Naïve Bayes Classifier.

**Figure 11 diagnostics-13-00739-f011:**
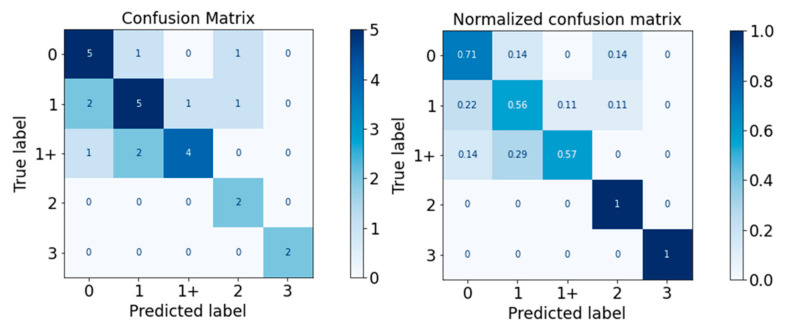
Confusion Matrix of Decision Tree Classifier.

**Figure 12 diagnostics-13-00739-f012:**
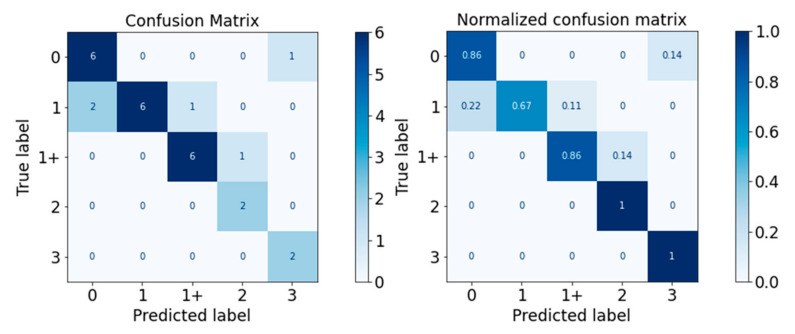
Confusion Matrix of Random Forest Classifier.

**Figure 13 diagnostics-13-00739-f013:**
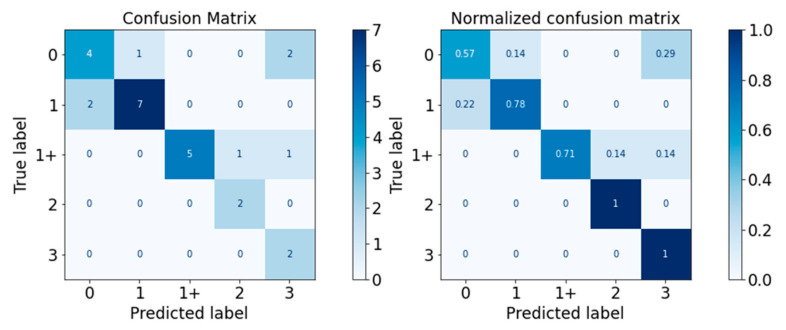
Confusion Matrix of XGBoost Classifier.

**Figure 14 diagnostics-13-00739-f014:**
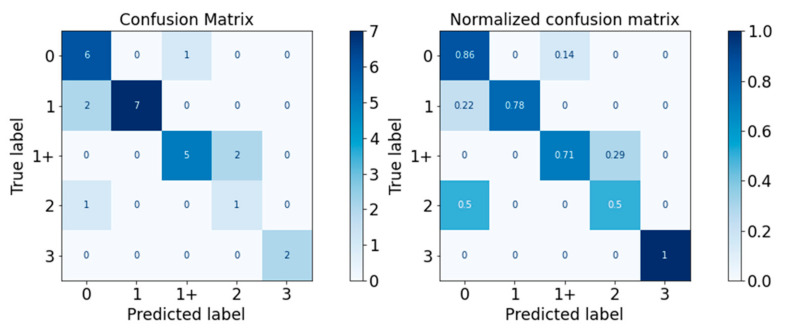
Confusion Matrix of SVM Classifier.

**Figure 15 diagnostics-13-00739-f015:**
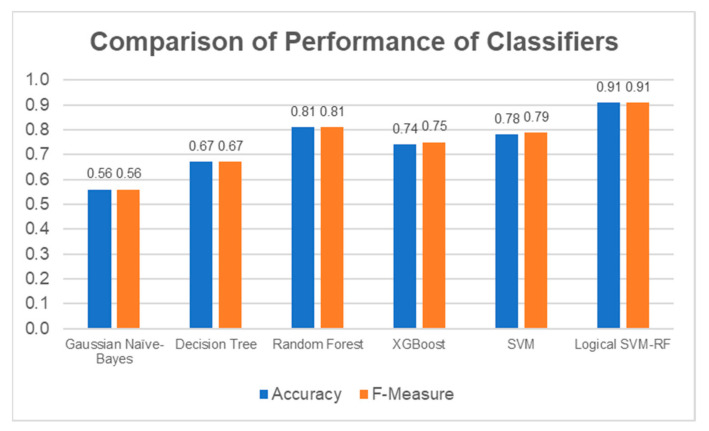
Accuracy and F-Measure Scores of Each Classifier.

**Figure 16 diagnostics-13-00739-f016:**
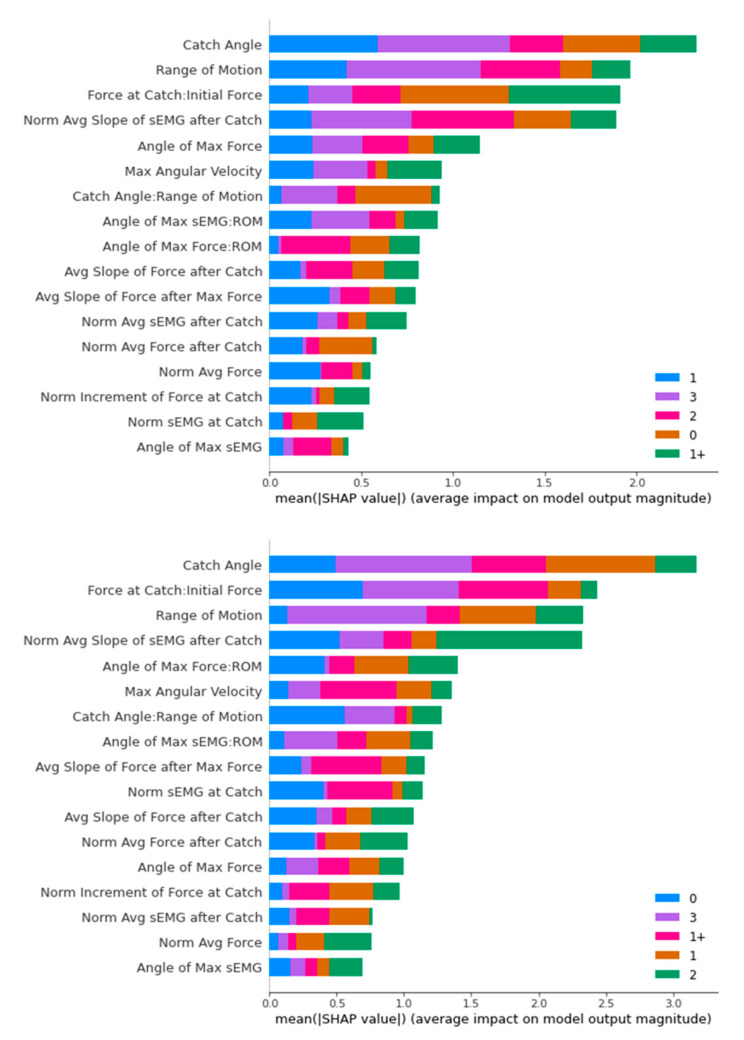
SHAP Analysis Result for Decision Tree: ROS-upsampled (**above**) and SMOTE-upsampled (**below**).

**Figure 17 diagnostics-13-00739-f017:**
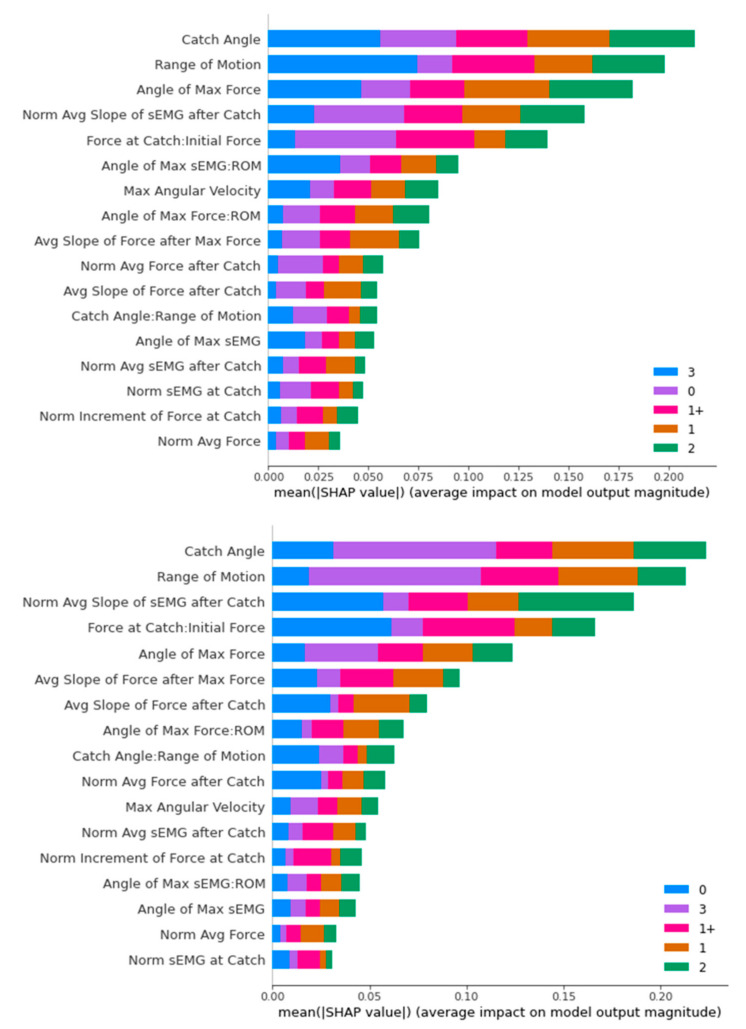
SHAP Analysis Result for Random Forest: ROS-upsampled (**above**) and SMOTE-upsampled (**below**).

**Figure 18 diagnostics-13-00739-f018:**
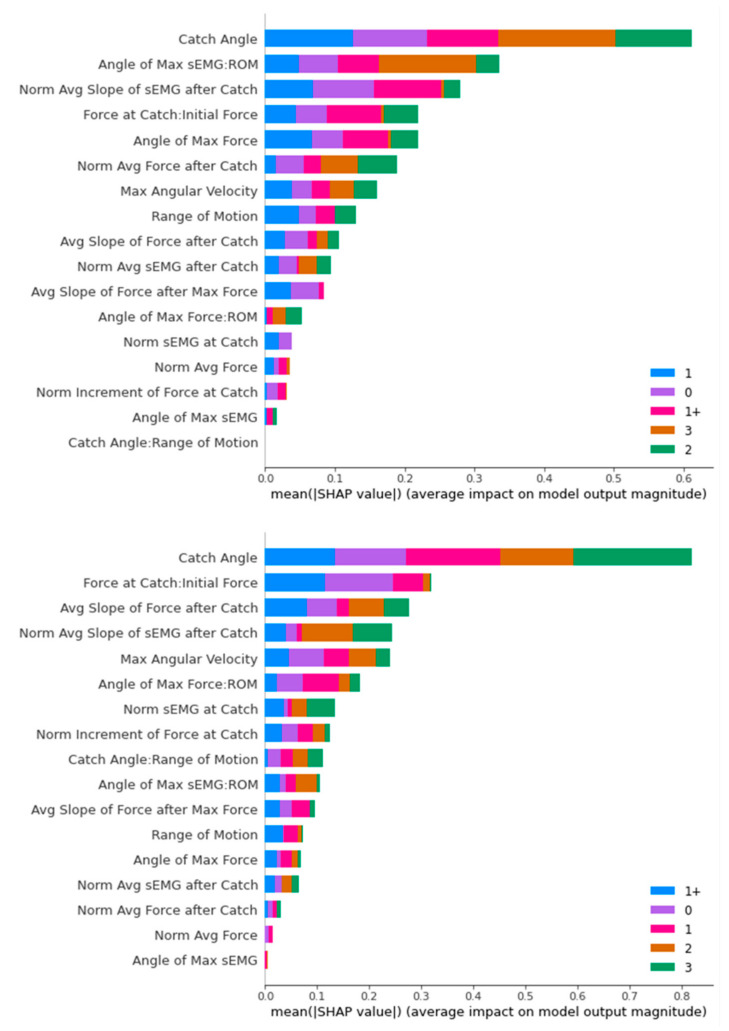
SHAP Analysis Result for XGBoost: ROS-upsampled (**above**) and SMOTE-upsampled (**below**).

**Figure 19 diagnostics-13-00739-f019:**
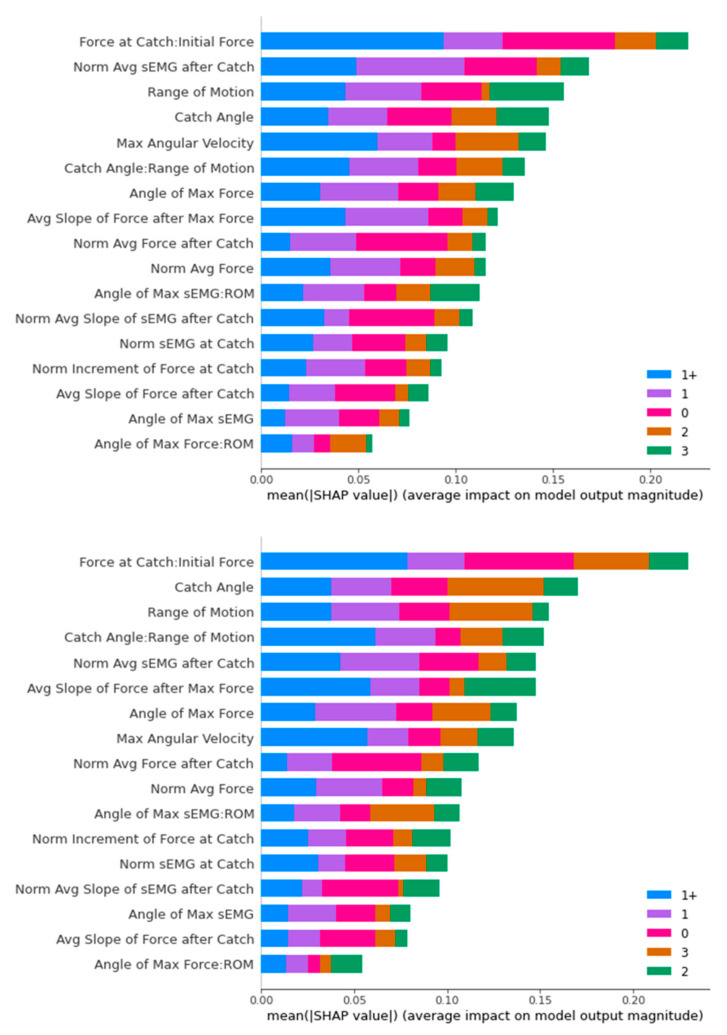
SHAP Analysis Result for SVM: ROS-upsampled (**above**) and SMOTE-upsampled (**below**).

**Figure 20 diagnostics-13-00739-f020:**
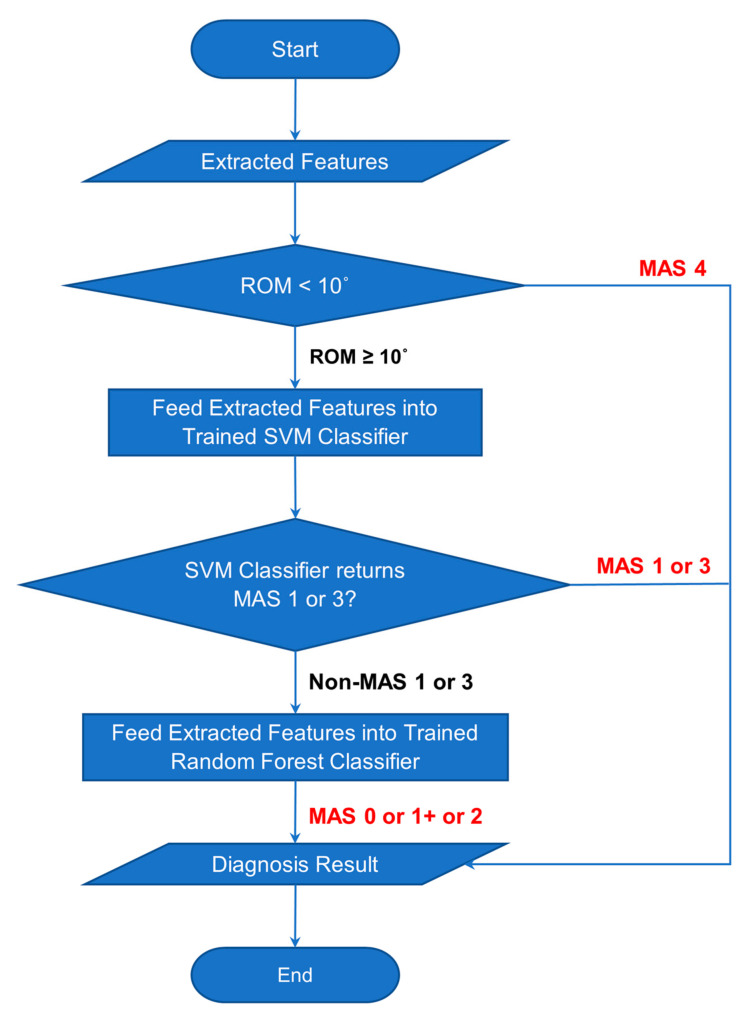
Flowchart of Classification Process of Combined Classifier.

**Figure 21 diagnostics-13-00739-f021:**
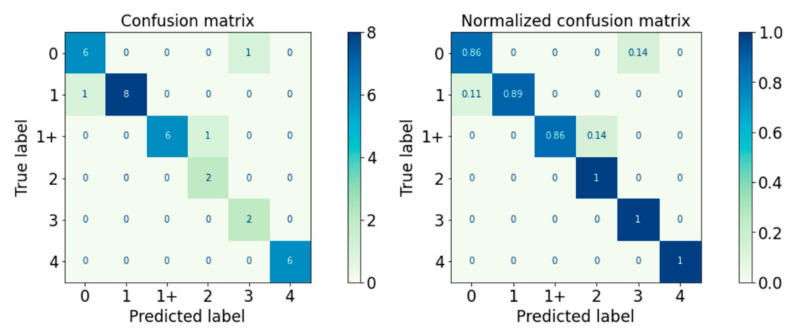
Confusion Matrix of Logical–SVM–RF Classifier.

**Figure 22 diagnostics-13-00739-f022:**
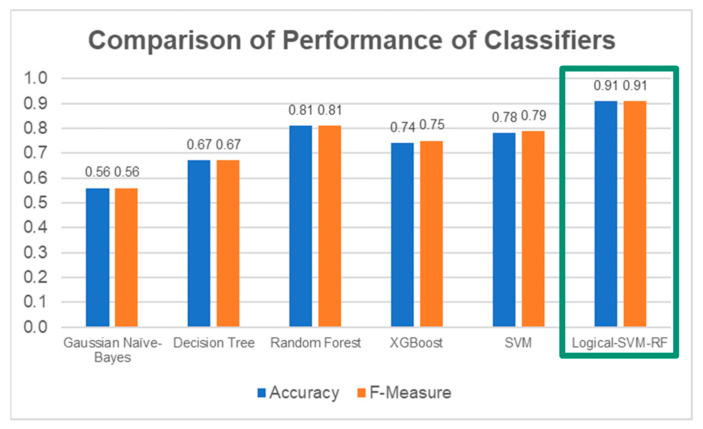
Comparison of performance between the individual classifiers and the Logical–SVM–RF classifier. The performance of Logical-SVM-RF classifier (in green box) is the highest among all.

**Figure 23 diagnostics-13-00739-f023:**
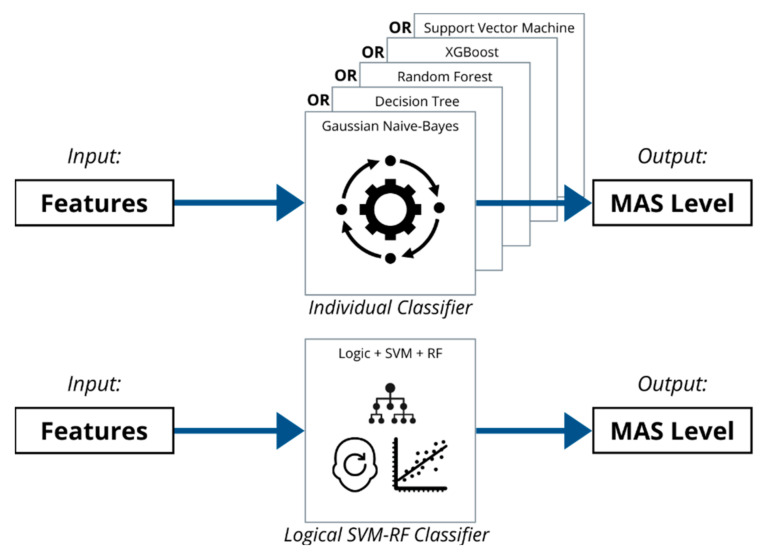
Logical–SVM–RF Classifier can better classify the MAS level based on the extracted features than the individual classifiers working alone.

**Table 1 diagnostics-13-00739-t001:** Visualisation of MAS Grades.

MAS Grades		
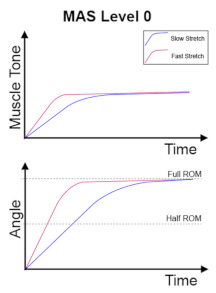	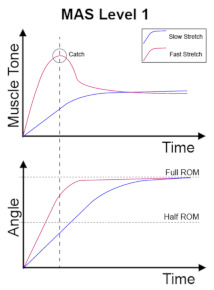	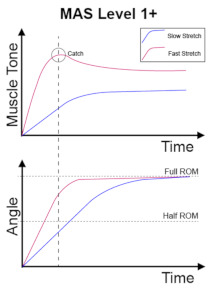
No increase in muscle tone.	Slight increase in muscle tone, manifested by a catch and release or by minimal resistance at the end of the range of motion when the affected part(s) is moved in flexion or extension.	Slight increase in muscle tone, manifested by a catch, followed by minimal resistance throughout the remainder (less than half) of the ROM.
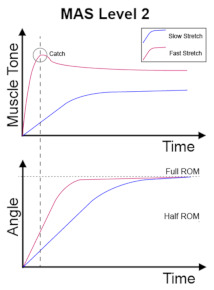	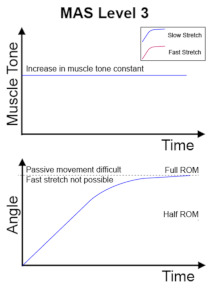	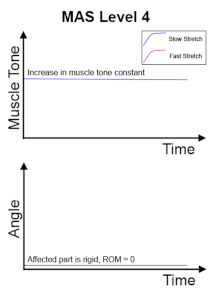
More marked increase in muscle tone through most of the ROM, but affected part(s) easily moved.	Considerable increase in muscle tone, passive movement difficult.	Affected part(s) rigid in flexion or extension.

**Table 2 diagnostics-13-00739-t002:** Sensors in Data Acquisition System.

Sensor	Specifications	
DataLITE Wireless Twin-Axis Goniometers	Range	0–340° (±170°)
	Accuracy	±2° measured over a range of ±90°
	Resolution	+0.1° in a range of 180°
	OperatingTemperature	+10 to +40 [°C]
DataLITE Handheld Myometer	Rated Load	0 to 50 [kg] (for compression only)
	Accuracy	Better than 1% rated load
DataLITE Wireless Surface EMG Sensor	Bandwidth	10–490 [Hz]
	Amplifier	Standard Unit × 1000
	Accuracy	±1.0 [%]
	Noise	<5 [µV]
	Sampling Rate	Up to 2000 [Hz]

**Table 3 diagnostics-13-00739-t003:** Distributions of Subjects’ MAS Level.

MAS Level	Number of Trials
0	28
1	28
1+	22
2	9
3	7
4	2
TOTAL	96

**Table 4 diagnostics-13-00739-t004:** Details of Sensors Data.

Sensors	Data	Unit
Twin-Axis Electrogoniometer	Elbow angle (x-axis)	Degree [°]
	Elbow angle (y-axis)	Degree [°]
Handheld Myometer	Elbow resisting force	Newton [N]
Wireless sEMG Sensor	Surface EMG	Millivolt [mV]

**Table 5 diagnostics-13-00739-t005:** Filter and Cleaning Processes for Different Data.

Data	Filter/Cleaning Processes
Elbow Angle	Median Filter (5th order)
	Mean Filter (5th order)
Elbow Resistance	Median Filter (5th order)
	Mean Filter (5th order)
	Fix Zero Data Levelling
Surface Electromyography	Zero Mean Value Data Levelling
	Value Rectification
	Butterworth Filter (4th order, 10 Hz Low-Pass Filter)

**Table 6 diagnostics-13-00739-t006:** Distributions of Datasets According to MAS Levels.

MAS Level	Train Set	Test Set	Total
0	65	7	72
1	75	9	84
1+	59	7	66
2	25	2	27
3	19	2	21
4	5	1	6
Total	248	28	276

**Table 7 diagnostics-13-00739-t007:** Grid search value range of each hyperparameter.

Classifier Type	Hyperparameter	Value Range
Gaussian Naïve Bayes	Variance Smoothing	From 10^−9^ to 1
Decision Tree	Criterion	[Gini, Entropy]
	Min Samples in Leaf	From 2 to 20
	Min Samples for Split	From 2 to 20
	Splitter	[Best, Random]
	Max Feature	From 2 to 10
Random Forest	Criterion	[Gini, Entropy]
	Min Samples in Leaf	From 2 to 20
	Min Samples for Split	From 2 to 20
	Splitter	[Best, Random]
	Max Feature	From 2 to 10
	N Estimators	[25, 50, 75, 100]
XGBoost	Booster	[gbtree]
	Gamma	[0.5, 1.5, 2, 5]
	Learning Rate	[0.01, 0.05, 0.1, 0.5]
	Min Child Weight	[5, 10]
	N Estimators	[50, 100, 200]
SVM	Kernel	[Linear, Radial Basis Function]
	Gamma	Scale, Auto
	C	0.01–100

**Table 8 diagnostics-13-00739-t008:** Optimised Hyperparameters for Gaussian Naïve Bayes.

Hyperparameter	Value	
	ROS	SMOTE
Variance Smoothing	0.1203	0.1203

**Table 9 diagnostics-13-00739-t009:** Optimised Hyperparameters for Decision Tree.

Hyperparameter	Value	
	ROS	SMOTE
Criterion	Entropy	Entropy
Min Samples in Leaf	3	3
Min Samples for Split	6	6
Splitter	Best	Random
Max Feature	8	8

**Table 10 diagnostics-13-00739-t010:** Optimised Hyperparameters for Random Forest.

Hyperparameter	Value	
	ROS	SMOTE
Criterion	Gini	Entropy
Min Samples in Leaf	3	6
Min Samples for Split	6	18
Splitter	Best	Best
Max Feature	3	4
N Estimators	25	25

**Table 11 diagnostics-13-00739-t011:** Optimised Hyperparameters for XGBoost.

Hyperparameter	Value	
	ROS	SMOTE
Booster	gbtree	gbtree
Gamma	0.1	0.1
Learning Rate	0.1	0.1
Min Child Weight	5	5
N Estimators	100	100
Objective	Multiclass:softprob	Multiclass:softprob

**Table 12 diagnostics-13-00739-t012:** Optimised Hyperparameters for Support Vector Machine

Hyperparameter	Value	
	ROS	SMOTE
C	15	15
Gamma	Gamma	Scale
Kernel	RBF	RBF

**Table 13 diagnostics-13-00739-t013:** Performance of Classifiers on Train Set.

Classifier	Accuracy ± Standard Deviation	
	ROS	SMOTE
Gaussian Naïve Bayes	0.59 ± 0.11	0.60 ± 0.16
Decision Tree	0.71 ± 0.16	0.60 ± 0.16
Random Forest	0.83 ± 0.13	0.72 ± 0.16
XGBoost	0.79 ± 0.18	0.75 ± 0.17
SVM	0.83 ± 0.14	0.79 ± 0.13

**Table 14 diagnostics-13-00739-t014:** Performance of Gaussian Naïve Bayes Classifier.

MAS Level	Precision	Recall	F-Measure
0	0.50	0.43	0.46
1	0.86	0.67	0.75
1+	1.00	0.29	0.44
2	0.25	1.00	0.40
3	0.50	1.00	0.67
Accuracy	-	-	0.56
Weighted Average	0.73	0.56	0.56

**Table 15 diagnostics-13-00739-t015:** Performance of Decision Tree Classifier.

MAS Level	Precision	Recall	F-Measure
0	0.63	0.71	0.67
1	0.63	0.56	0.59
1+	0.80	0.57	0.67
2	0.50	1.00	0.67
3	1.00	1.00	1.00
Accuracy	-	-	0.67
Weighted Average	0.69	0.67	0.67

**Table 16 diagnostics-13-00739-t016:** Performance of Random Forest Classifier.

MAS Level	Precision	Recall	F-Measure
0	0.75	0.86	0.80
1	1.00	0.67	0.80
1+	0.86	0.86	0.86
2	0.67	1.00	0.80
3	0.67	1.00	0.80
Accuracy	-	-	0.81
Weighted Average	0.85	0.81	0.81

**Table 17 diagnostics-13-00739-t017:** Performance of XGBoost Classifier.

MAS Level	Precision	Recall	F-Measure
0	0.67	0.57	0.62
1	0.88	0.78	0.82
1+	1.00	0.71	0.83
2	0.67	1.00	0.80
3	0.40	1.00	0.57
Accuracy	-	-	0.74
Weighted Average	0.80	0.74	0.75

**Table 18 diagnostics-13-00739-t018:** Performance of SVM Classifier.

MAS Level	Precision	Recall	F-Measure
0	0.67	0.86	0.75
1	1.00	0.78	0.88
1+	0.83	0.71	0.77
2	0.33	0.50	0.40
3	1.00	1.00	1.00
Accuracy	-	-	0.78
Weighted Average	0.82	0.78	0.79

**Table 19 diagnostics-13-00739-t019:** Performance of Logical–SVM–RF Classifier.

MAS Level	Precision	Recall	F-Measure
0	0.86	0.86	0.86
1	1.00	0.89	0.94
1+	1.00	0.86	0.92
2	0.67	1.00	0.80
3	0.67	1.00	0.80
4	1.00	1.00	1.00
Accuracy	-	-	0.91
Weighted Average	0.93	0.91	0.91

**Table 20 diagnostics-13-00739-t020:** Comparison of Logical–SVM–RF classifier and other existing works.

Aspect	Yee et al. (This Work)	Ahmad Puzi et al. [6]	Park et al. [7]	Zhang et al. [8]	Chen et al. [9]
Stretching Method	Passive	Passive	Passive	Passive	Active
Clinical Scale	MAS 0, 1, 1+, 2, 3, 4	MAS 0, 1, 2	MAS 0, 1, 1+, 2, 3	MAS 0, 1, 1+, 2, 3	MAS 0, 1, 1+, 2
MAS Classes	6	3	5	5	4
No. of Subjects	50	25	34	24	13
Sensors Data	Angle, Force, sEMG	Angle, Torque	Angle, Torque	IMU, sEMG	IMU, sEMG
Classification Methods	SVM + RF + Logical Decision	SVM, Linear Discriminant, Weighted-KNN	MLP	Support Vector Regressor	KNN, SVM, RF, MLP
Reported Performance	Accuracy & F1: 91%	Accuracy: 76–84%	Accuracy: 82.2%	MSE: 0.059	F1: 70.5–95.2%

**Table 21 diagnostics-13-00739-t021:** Comparison of current clinical practice and assisted diagnosis system.

Aspect	Conventional Clinical Practice	Assisted Diagnosis System
Setup time	No setup required	<5 min
Measurement Data	Elbow Angle	Elbow Angle, Elbow Resisting Force, sEMG
Data Format	Recorded with pen and paper	Recorded as numerical data in digital format
Diagnosis Method	Manual	Assisted by data-driven machine learning model
Diagnosis Outcome	Inter-rater and intra-rater variability issues	Transparency in decision making

## Data Availability

The data presented in this study are available on request from the corresponding author.

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
