# Peer review of "Clinical Spasticity Assessment Assisted by Machine Learning Methods and Rule-Based Decision"

_diagnostics, 2023, doi:10.3390/diagnostics13040739_

Round 1

Reviewer 1 Report

Grammar and Readability: The paper is well-written and clear. I didn't find any typos.

Minor review

In this manuscript, the authors provide an experimental evaluation of different ML models for the detection and classification related to the problem of spasticity. In particular, the authors define an interesting ML model that combines several simple models to increase the performance of these. Both sections talk about the data preparation process, and the section where the new model is illustrated is interesting and provides some food for thought to the reader. Moreover, the paper is clear and well-written. 

Although this work is interesting, there are some issues that the authors should solve:

- The abstract does not highlight the specifics of the research or findings but contains too much background information. Some details of the research would be nice for example numbers addressing the sample, data, percentage improvement, etc.. Remove some of the background material and add some details of the research. Moreover, it is good to provide some specifics (e.g., sample size, dataset size, numbers from results, etc.).  

- Related work section is missing. Please add this section also adding more recent references in the area. Certainly, there has been more recent (within the last two years) research on this topic published in information science and/or computer science outlets. An academic search on the topic (using keywords from the manuscript’s title) shows that there is recent work in this area. Therefore, I suggest some works start with: https://doi.org/10.1007/s40192-021-00243-2, https://doi.org/10.1145/3487664.3487719, https://doi.org/10.1007/978-3-662-62271-1_5 

- In this work, there is no comparative evaluation between the proposed model and other models existing in the literature. I suggest adding this comparative evaluation to show if this proposal outperforms other proposals.

- In this case, I suggest describing more works existing in the literature. Moreover, I suggest creating a new section for this, removing the relative part from the introduction and enhancing the latter.

- Other general issues:

 - remove the repetition of "Figure 1" at line 51.

 - in table 2, there are 3 columns but only two labels. please, add the missing label

 - The quality of Figure 7 is poor. I suggest improving it.

 - The header of Table 6 is split from the rest of the table. Please, fix it.

 - sections 5.3.1 and 6.2.1 have the same title (same for sections 5.3.2 and 6.2.2). This can confuse the reader; I suggest changing the titles or merging the subsections.

 - At line 476, there is an error message. Please, remove it.

 - The quality of the figures should be improved.

Concluding Remarks:

The goal of the paper is interesting. I think that the paper could be improved with the considerations I reported in the review.

Author Response

Dear Reviewer,

Thank you for taking your precious time in reviewing the paper and provided us with your valuable comments. We have carefully considered the given comments, and the best effort has been put in to address every one of them. We sincerely hope the revised manuscript meet your standard. Please see the attachment for the response to your review.

Best regards,

Cheng Yee Low

Reviewer 2 Report

As MAS can be subjective due to its quantitative nature, the manuscript concerns the clinical assessment of spasticity using machine learning. The methods were developed from data acquired from 50 subjects. 

- Line 14 - 27: Please include the numerical results in the abstract.

- Lines 91 - 96: In Section 2.1, please present the inclusion and exclusion criteria as a list or table. 

- Lines 91 - 96: Did you obtain informed consent from the patients?

- Lines 101 - 111: Please provide the time taken for each process and the background

- Lines 101 - 111: Please provide background and expertise of the physicians who were part of the research.

- Lines 113 - 118: Please include the brand names and models of the devices used in the study in Table 2.

- Lines 192 - 197: Please provide the patient characteristics as per the inclusion criteria. .

- Lines 213 - 216: How do you perform time and frequency synchronization of the data acquired from different sensors?

- Lines 233 - 243: What are the specification of each filter that you used?

- Line 235: What is the value of k?

- Section 3, 4, and 5 should be integrated into Section 2 Materials and Methods.

- Line 365: Are [Line 235 = stratified k-fold splitting] and [Line 365/374 = 10-fold cross-evaluation] the same thing?

- Lines 421 - 433: You did not write down the range of parameters that you used for optimization. Pplease provide the range of parameters used for optimization in Lines 367 - 372.

- Line 518: Generally XGBoost and RF should have similar performances, but the gaps are quite high. This may be due to the limited number of data.

- Section 7: Please include more information about how this research can be implemented in clinical practice (if possible) and the advantages of implementing it, as well as the future direction of the work.

Author Response

Dear Reviewer.

Thank you for taking your precious time to review the paper and providing us with your valuable comments. We have carefully considered the given comments, and the best effort has been put in to address every one of them. We sincerely hope the revised manuscript meets your standard. Please see the attachment for the response to your review.

Best regards,

Cheng Yee Low

Reviewer 3 Report

This work is a clinical assessment , classification and feature selection of spasticity using quantitative measurement data by wearable sensors. Work is good. Some minors update can improved the paper more for publication.

1. The use of English has to improve.There are many grammatical and syntax errors.

2. Introduction has well conducted. The contribution is clear which should be conveyed through the paper.

3. Discussion section should be some comparative table and figures to show and compare work with existing works.

Author Response

Dear Reviewer,

Thank you for taking your precious time to review the paper and providing us with your valuable comments. We have carefully considered the given comments, and the best effort has been put in to address every one of them. We sincerely hope the revised manuscript meets your standard. Please see the attachment for the response to your review.

Best regards,

Cheng Yee Low
